# Research on the Multi-Point Gravity Balance Method for the Large-Aperture Space Camera in the Imaging Quality Test

**Lihao Zhang \*, Mingxin An, Weilu Wang and Xiaobo Li**

Changchun Institute of Optics, Fine Mechanics and Physics, Chinese Academy of Sciences, Changchun 130033, China
\* Correspondence: zhanglihao@ciomp.ac.cn; Tel.: +86-135-0088-6653

**Abstract:** The large-aperture space camera needs gravity unloading in the ground imaging quality test. The multi-point gravity balance method is mainly applied to the object with symmetrical structure and regular mass distribution, such as the primary mirror of the space camera. There is no relevant research on whether large-aperture space cameras with large weight and low stiffness can be unloaded by a multi-point gravity balance system. Based on the equivalent statically determinate structure and the classical equation of force method, the load-solving method of a multi-point gravity balance system is studied. The mathematical model of a two-point unit and a three-point unit is established with the principle of force balance. The mathematical model of a gravity balance system is built with the unit-grouping principle. A 15-point gravity balance system of a large-aperture space camera is established. The displacements of the mirrors under the multi-point gravity balance system are much smaller than the displacements under the traditional support by the finite element analysis, which can realize the imaging quality test. The multi-point gravity balance method is effective on the space camera imaging quality test.

**Keywords:** large aperture; space camera; imaging quality test; gravity unloading; gravity balance





## 1. Introduction

The space camera needs to simulate the on-orbit working state on the ground and conduct sufficient imaging quality tests. The large-aperture space camera has a large structure size, large weight and complex distribution, and low structural stiffness. During the horizontal state test of optical axis, the test system only supports the mounting surface of the space camera and satellite platform. The displacements of the mirrors reach the millimeter level by gravity. It is impossible to realize the imaging quality test. Therefore, the imaging quality test of the large-aperture space camera requires a gravity unloading device to restore the displacement of the mirrors to the micron level.

The commonly used gravity compensation methods are the floating method, magnetic levitation method, suspension method and balance method. The floating method cannot be used in the vacuum test environment. The magnetic levitation method and suspension method need to design a set of independent loading systems [1–4]. When there are many gravity unloading points required, the system of the magnetic levitation method and suspension method is extremely complicated. The balance method adopts the whiffletree structure to realize the gravity unloading by the own gravity of the object. Compared with the above gravity compensation methods, the system composition of the balance method is simple, suitable for the vacuum test environment, and no external force is introduced for gravity unloading.

The multi-point gravity balance method is mainly applied to the object with symmetrical structure and regular mass distribution. The support of the primary mirror of the VLTI telescope is a 54-point whiffletree structure [5]. The support of the primary mirror of the SOFIA telescope is an 18-point whiffletree support structure, as shown in Figure 1 [6].

The support of the mirror of the European Extremely Large Telescope E-ELT is a 27-point whiffletree support structure [7–9]. The support of the mirror of the TMT telescope is a 27-point whiffletree support structure [10–15]. There is no relevant research on whether large-aperture space cameras with large weight and low stiffness can be unloaded by a multi-point gravity balance system [16–23].

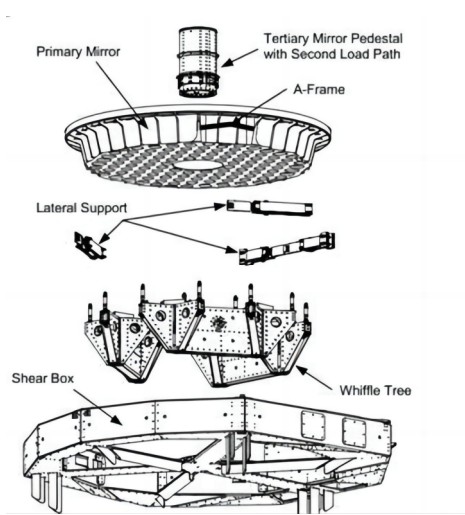

**Figure 1.** The whiffletree support structure of the SOFIA telescope.

This paper mainly studies the mechanism of a multi-point gravity balance system and the mathematical model of gravity unloading for asymmetric structures with low-stiffness mass distribution. The gravity balance system of a large-aperture space camera is established to meet the requirements of the imaging quality test on the ground.

## 2. Study on Mathematical Model of Multi-Point Gravity Balancing Method

In order to realize the imaging quality test of the large-aperture space camera on the ground, it is necessary to apply multiple gravity unloading points at the key positions of the main structure to compensate for the gravity deformation of the main structure. The key to establishing the mathematical model of the gravity balance method is obtaining the load of each gravity unloading point of the first stage. According to the principle of force balance, a single-stage gravity balance system model is obtained. A single stage can reduce two or three unloading points to one point. Finally, it is equivalent to three unloading points, and the whole system model is obtained. The schematic diagram of the gravity balance system of a large-aperture space camera is shown in Figure 2.

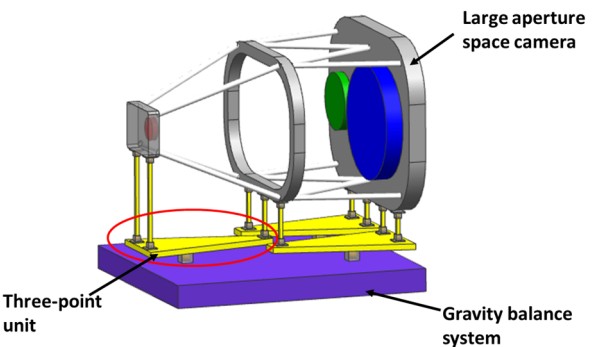

**Figure 2.** Schematic diagram of the gravity balance system of a large-aperture space camera.

### 2.1. Multi-Point Load-Solving Method

The mathematical model of a gravity balance system needs to solve the load at each gravity unloading point of the first stage. The mass distribution of the large-aperture space

camera is uneven. The unloading point is distributed due to the inconsistent stiffness of the camera structure. The unloading point load cannot be solved simply. The number of unloading points in the multi-point gravity balance method is $m$ ($m > 3$), which is more than the number of constraint equations in the whole system. The unloading point load cannot be directly derived. If the three unloading points load are replaced by three gravity direction displacement constraints, the gravity balance system becomes a statically determinate structure and $n$ ($n = m - 3$) unknown forces. The displacement of $n$ points is zero under $n$ unknown forces, which is the same as the original structure. N displacements and forces equations are defined as follows:

$$\delta_{11}F_1 + \delta_{12}F_2 + \ldots + \delta_{1n}F_n + \Delta_{1P} = 0$$
$$\cdots\cdots$$
$$\cdots\cdots$$
$$\delta_{n1}F_1 + \delta_{n2}F_2 + \ldots + \delta_{nn}F_n + \Delta_{nP} = 0 \tag{1}$$

$\delta ii$ is the main coefficient, the displacement of point $i$ when point $i$ is under a unit force (Fi = 1), which is always positive.

$$\delta_{ii} = \sum \int \frac{\overline{M_i^2}\mathrm{ds}}{EI} + \sum \int \frac{\overline{N_i^2}\mathrm{ds}}{EA} + \sum \int u\frac{\overline{Q_i^2}\mathrm{ds}}{GA} \tag{2}$$

$\delta ij$ is the coupling coefficient, the displacement of point $j$ when point $i$ is under a unit force (Fi = 1), which can be positive, negative or zero.

$$\delta_{ij} = \sum \int \frac{\overline{M_iM_j}\mathrm{ds}}{EI} + \sum \int \frac{\overline{N_iN_j}\mathrm{ds}}{EA} + \sum \int u\frac{\overline{Q_iQ_j}\mathrm{ds}}{GA} \tag{3}$$

$\Delta ip$ is a free term, the displacement of point $i$ when the structure is under a gravity load, which can be positive, negative or zero.

$$\Delta_{iP} = \sum \int \frac{\overline{M_iM_P}\mathrm{ds}}{EI} + \sum \int \frac{\overline{N_iN_P}\mathrm{ds}}{EA} + \sum \int u\frac{\overline{Q_iQ_P}\mathrm{ds}}{GA} \tag{4}$$

N unknown forces can be solved with $n$ equations. The three direction displacement constraints are replaced by three gravity unloading points loads. The force balance equations of a statically determinate structure are Equations (5)–(7). The other three forces can be solved by the three equations. The loads of all $m$ points are solved.

$$F_1(x_1 - x_G) + F_2(x_2 - x_G) + \cdots F_m(x_m - x_G) = 0 \tag{5}$$

$$F_1(y_1 - y_G) + F_2(y_2 - y_G) + \cdots F_m(y_m - y_G) = 0 \tag{6}$$

$$F_1 + F_2 + \cdots F_m + G = 0 \tag{7}$$

### 2.2. Mathematical Model of Balance System

The loads at all the unloading points of the first stage are solved in the previous section. Based on the principle of force balance, the functional relationship between the balance location and the load of a three-point unit or two-point unit is deduced in this section.

### 2.2.1. The Three-Point Unit

The three-point unit equates three point forces to a single point force. There is a balance point on the plane determined by the three forces, as shown in Figure 3. Three equations can be obtained according to the principle of force balance, as follows:

$$F_1(x_1 - x) + F_2(x_2 - x) + F_3(x_3 - x) = 0 \tag{8}$$

$$F_1(y_1 - y) + F_2(y_2 - y) + F_3(y_3 - y) = 0 \tag{9}$$

$$F' + F_1 + F_2 + F_3 = 0 \tag{10}$$

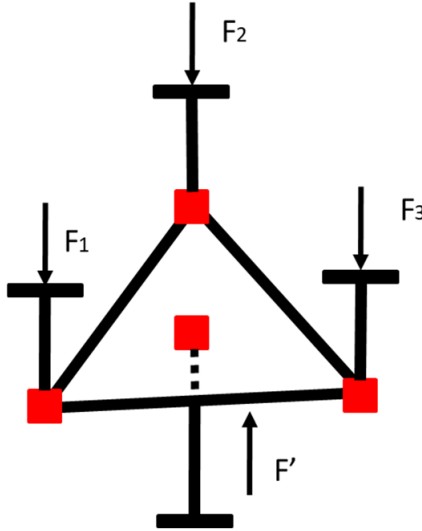

**Figure 3.** Schematic diagram of three-point unit.

Through the above three equations, the location of the balance point and the input force of the next stage of the balance structure can be obtained.

$$x = \frac{F_1 x_1 + F_2 x_2 + F_3 x_3}{F_1 + F_2 + F_3} \tag{11}$$

$$y = \frac{F_1 y_1 + F_2 y_2 + F_3 y_3}{F_1 + F_2 + F_3} \tag{12}$$

$$F' = -(F_1 + F_2 + F_3) \tag{13}$$

2.2.2. The Two-Point Unit

The two-point unit equates two point forces to a single point force. There is a balance point on the line determined by the two forces, as shown in Figure 4. Three equations can be obtained according to the principle of force balance, as follows:

$$F_1(x_1 - x) + F_2(x_2 - x) = 0 \tag{14}$$

$$F_1(y_1 - y) + F_2(y_2 - y) = 0 \tag{15}$$

$$F' + F_1 + F_2 = 0 \tag{16}$$

Through the above three equations, the location of the balance point and the input force of the next stage of the balance structure can be obtained.

$$x = \frac{F_1 x_1 + F_2 x_2}{F_1 + F_2} \tag{17}$$

$$y = \frac{F_1 y_1 + F_2 y_2}{F_1 + F_2} \tag{18}$$

$$F' = -(F_1 + F_2) \tag{19}$$

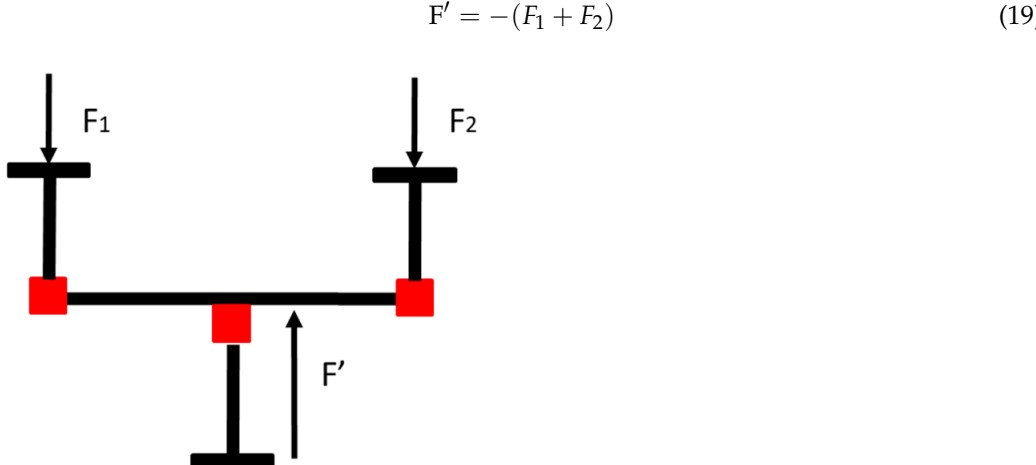

**Figure 4.** Schematic diagram of two-point unit.

### 2.2.3. The Balance System

The single stage of the gravity balance system is realized by the combination of a two-point unit or three-point unit, as shown in Figure 5. The number of multi-point gravity balance points $n_0$ is replaced by $a_1$ three-point units and $b_1$ two-point units, and the number of support points in the next stage is reduced $(2a_1 + b_1)$. The number of support points of the second stage is $(a_1 + b_1)$, and the number of support points can be reduced $(2a_2 + b_2)$, when these balance points are combined with two- or three-point units to achieve the second-stage configuration. Based on this principle, the number of balance points decreases step by step, and the basic principle of the gravity balance system is finally determined.

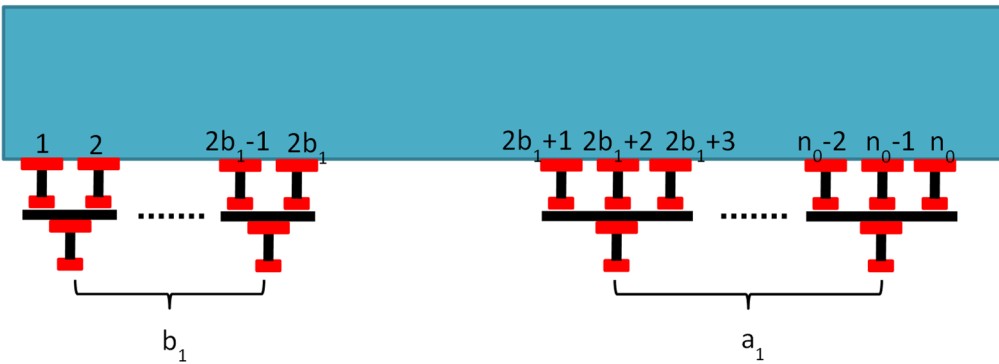

**Figure 5.** Single-stage schematic diagram.

For example, when the gravity balance point is 31, the gravity balance system is divided into three levels, consisting of 18 units. The first level is composed of 10 three-point units and a single fulcrum, and the number of fulcrums is reduced from 31 to 11. The second stage consists of one two-point unit and three three-point units, and the number of fulcrums is reduced from 11 to 4. The final stage consists of a two-point unit and a single fulcrum, which is ultimately equivalent to a three-point statically indeterminate system.

The multiple units are grouped to form the balance system. The grouping principle is as follows: Multiple points should be grouped according to the shortest distance to reduce the size of a single balance unit. The balance point is considered comprehensively to avoid crossing between different unit structures. It is more beneficial to reduce the scale of the gravity balancing system to adopt more three-point units. According to the above grouping principles, the function relationship between the balance location and the balance force of each stage is derived step by step, and the mathematical model of the system is finally obtained.

## 3. Multi-Point Gravity Balance System Model for a Large-Aperture Space Camera

### 3.1. Multi-Point Load of the First Stage of the Gravity Balance System

A large-aperture space camera is an off-axis three-mirror optical system equipped with a variety of scientific instruments. The space camera weighs 5 tons and is about 10 m long and 5 m wide. The main structure is a four-layer frame to ensure the structural stability of the large-aperture optical system. The primary mirror and the tertiary mirror are installed on the main frame, and the secondary mirror is installed on the front frame. Some scientific instruments are arranged on the front frame and the middle frame, limited by the envelope of the launch vehicle. The model of a large-aperture space camera is shown in Figure 6.

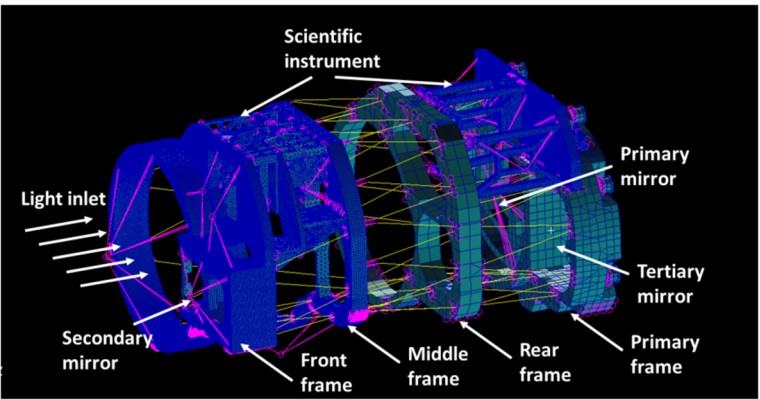

**Figure 6.** Large-aperture space camera.

The main frame is the basis of the entire camera and carries the primary mirror, the tertiary mirror and some scientific instruments. Five unloading points are selected according to the mass distribution. The front frame carries some scientific instruments and the secondary mirror. According to the mass distribution, four unloading points are selected. The middle two frames carry the weight of some of the scientific instruments, and only three unloading points are required. According to mass distribution and the four-layer frame structure form of the large-aperture space camera, 15 unloading points are preliminarily determined for gravity unloading. The locations of the 15 points are shown in Figure 7, and the location coordinates are shown in Table 1.

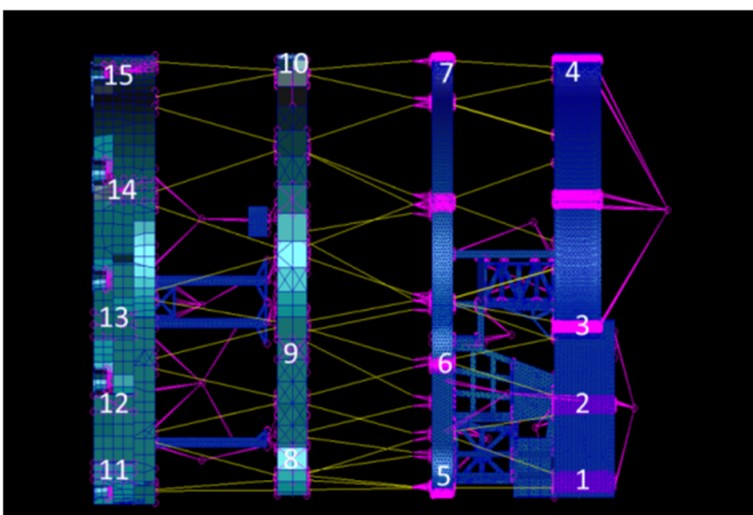

**Figure 7.** The distribution of gravity balance point of a large-aperture space camera. Point 1, 2, 3, 4 on the front frame; Point 5, 6, 7 on the first middle frame; Point 8,9,10 on the second middle frame; Point 11, 12, 13, 14, 15 on the main frame.

**Table 1.** The location coordinates of 15 unloading points.

| Unloading Point | X (m) | Y (m) | Z (m) |
|---|---|---|---|
| 1 | 0.490 | −0.701 | −4.220 |
| 2 | 0.490 | −0.061 | −4.220 |
| 3 | 0.693 | 0.590 | −4.154 |
| 4 | 0.381 | 2.775 | −4.154 |
| 5 | 0.308 | −0.780 | −2.862 |
| 6 | 0.550 | 0.278 | −2.868 |
| 7 | 0.256 | 2.793 | −2.861 |
| 8 | 0.673 | −0.508 | −1.416 |
| 9 | 0.705 | 0.392 | −1.416 |
| 10 | 0.755 | 2.560 | −1.412 |
| 11 | 0.809 | −0.611 | 0.300 |
| 12 | 0.888 | −0.050 | 0.300 |
| 13 | 1.080 | 0.597 | 0.300 |
| 14 | 1.185 | 1.706 | 0.251 |
| 15 | 0.521 | 2.688 | 0.161 |

According to the multi-point load solution method in the last section, the three unloading points 3, 11 and 15 are replaced by three gravity direction displacement constraints. The gravity balance system becomes statically determinate.

(a) When point 1 loads a unit force, the displacement of point 1 is obtained by the finite element analysis. It is the main coefficient of point 1 $\delta_{11}$. Other points also use the above solution to obtain the main coefficient of the point.

(b) When point 1 loads a unit force, the displacement of the other point is obtained by the finite element analysis. It is the coupling coefficient of the point 1 $\delta_{1j}$. Other points also use the above solution to obtain the coupling coefficient of the point. The main coefficients and the coupling coefficients are shown in Table 2.

**Table 2.** The main coefficients and the coupling coefficients.

| $\delta_{ij}$ | 1 | 2 | 4 | 5 | 6 | 7 | 8 | 9 | 10 | 12 | 13 | 14 |
|---|---|---|---|---|---|---|---|---|---|---|---|---|
| 1 | $5.40 \times 10^{-8}$ | $3.03 \times 10^{-8}$ | $-1.12 \times 10^{-8}$ | $3.71 \times 10^{-8}$ | $2.07 \times 10^{-8}$ | $-9.63 \times 10^{-9}$ | $1.90 \times 10^{-8}$ | $1.19 \times 10^{-8}$ | $-5.09 \times 10^{-9}$ | $2.92 \times 10^{-11}$ | $-8.00 \times 10^{-11}$ | $-2.32 \times 10^{-10}$ |
| 2 | $3.03 \times 10^{-8}$ | $2.21 \times 10^{-8}$ | $-2.76 \times 10^{-9}$ | $2.11 \times 10^{-8}$ | $1.32 \times 10^{-8}$ | $-2.70 \times 10^{-9}$ | $1.09 \times 10^{-8}$ | $7.24 \times 10^{-9}$ | $-1.49 \times 10^{-0}$ | $-9.00 \times 10^{-12}$ | $-8.80 \times 10^{-11}$ | $-1.70 \times 10^{-10}$ |
| 4 | $-1.12 \times 10^{-9}$ | $-2.76 \times 10^{-9}$ | $1.13 \times 10^{-7}$ | $-1.02 \times 10^{-8}$ | $1.10 \times 10^{-8}$ | $7.38 \times 10^{-8}$ | $-3.68 \times 10^{-9}$ | $7.88 \times 10^{-8}$ | $3.74 \times 10^{-8}$ | $-4.64 \times 10^{-10}$ | $-7.99 \times 10^{-10}$ | $-4.92 \times 10^{-10}$ |
| 5 | $3.71 \times 10^{-8}$ | $2.11 \times 10^{-8}$ | $-1.02 \times 10^{-8}$ | $5.49 \times 10^{-8}$ | $2.96 \times 10^{-8}$ | $-7.79 \times 10^{-9}$ | $2.64 \times 10^{-8}$ | $1.71 \times 10^{-8}$ | $-2.94 \times 10^{-9}$ | $2.67 \times 10^{-10}$ | $2.65 \times 10^{-10}$ | $-2.20 \times 10^{-11}$ |
| 6 | $2.07 \times 10^{-8}$ | $1.32 \times 10^{-8}$ | $1.10 \times 10^{-8}$ | $2.96 \times 10^{-8}$ | $4.48 \times 10^{-8}$ | $9.46 \times 10^{-9}$ | $1.68 \times 10^{-8}$ | $1.44 \times 10^{-8}$ | $6.65 \times 10^{-9}$ | $1.51 \times 10^{-10}$ | $1.29 \times 10^{-10}$ | $-6.20 \times 10^{-11}$ |
| 7 | $-9.63 \times 10^{-9}$ | $-2.70 \times 10^{-9}$ | $7.38 \times 10^{-8}$ | $-7.79 \times 10^{-9}$ | $9.46 \times 10^{-9}$ | $7.42 \times 10^{-8}$ | $-1.52 \times 10^{-9}$ | $9.08 \times 10^{-0}$ | $3.75 \times 10^{-8}$ | $-2.28 \times 10^{-10}$ | $-3.78 \times 10^{-10}$ | $-2.39 \times 10^{-10}$ |
| 8 | $1.90 \times 10^{-8}$ | $1.09 \times 10^{-8}$ | $-3.68 \times 10^{-9}$ | $2.64 \times 10^{-8}$ | $1.68 \times 10^{-8}$ | $-1.52 \times 10^{-9}$ | $3.30 \times 10^{-8}$ | $2.21 \times 10^{-8}$ | $2.02 \times 10^{-9}$ | $5.01 \times 10^{-10}$ | $6.13 \times 10^{-10}$ | $1.93 \times 10^{-10}$ |
| 9 | $1.19 \times 10^{-8}$ | $7.24 \times 10^{-9}$ | $7.88 \times 10^{-9}$ | $1.71 \times 10^{-8}$ | $1.44 \times 10^{-8}$ | $9.08 \times 10^{-9}$ | $2.21 \times 10^{-8}$ | $2.41 \times 10^{-8}$ | $9.33 \times 10^{-9}$ | $4.88 \times 10^{-10}$ | $6.57 \times 10^{-10}$ | $2.41 \times 10^{-10}$ |
| 10 | $-5.09 \times 10^{-9}$ | $-1.49 \times 10^{-9}$ | $3.74 \times 10^{-8}$ | $-2.94 \times 10^{-9}$ | $6.65 \times 10^{-9}$ | $3.75 \times 10^{-8}$ | $2.02 \times 10^{-9}$ | $9.33 \times 10^{-9}$ | $3.93 \times 10^{-8}$ | $2.04 \times 10^{-11}$ | $6.28 \times 10^{-11}$ | $1.40 \times 10^{-10}$ |
| 12 | $2.92 \times 10^{-11}$ | $-9.00 \times 10^{-12}$ | $-4.64 \times 10^{-10}$ | $2.67 \times 10^{-10}$ | $1.51 \times 10^{-10}$ | $-2.28 \times 10^{-10}$ | $5.01 \times 10^{-10}$ | $4.88 \times 10^{-10}$ | $2.04 \times 10^{-11}$ | $2.68 \times 10^{-9}$ | $1.86 \times 10^{-9}$ | $6.18 \times 10^{-10}$ |
| 13 | $-8.00 \times 10^{-11}$ | $-8.80 \times 10^{-11}$ | $-7.99 \times 10^{-10}$ | $2.65 \times 10^{-10}$ | $1.29 \times 10^{-10}$ | $-3.78 \times 10^{-10}$ | $6.13 \times 10^{-10}$ | $6.57 \times 10^{-10}$ | $6.28 \times 10^{-11}$ | $1.86 \times 10^{-9}$ | $3.52 \times 10^{-9}$ | $1.33 \times 10^{-9}$ |
| 14 | $-2.32 \times 10^{-10}$ | $-1.70 \times 10^{-10}$ | $-4.92 \times 10^{-10}$ | $-2.20 \times 10^{-11}$ | $-6.20 \times 10^{-11}$ | $-2.39 \times 10^{-10}$ | $1.93 \times 10^{-10}$ | $2.41 \times 10^{-10}$ | $1.40 \times 10^{-10}$ | $6.18 \times 10^{-10}$ | $1.33 \times 10^{-9}$ | $4.22 \times 10^{-9}$ |

(c) The displacements of all points are obtained by the finite element analysis under gravity condition. They are the free terms $\Delta_{iP}$, as shown in Table 3.

**Table 3.** The free terms.

| | 1 | 2 | 4 | 5 | 6 | 7 | 8 | 9 | 10 | 12 | 13 | 14 |
|---|---|---|---|---|---|---|---|---|---|---|---|---|
| $\Delta_{iP}$ | $-3.04$ $\times 10^{-4}$ | $-3.30$ $\times 10^{-4}$ | $-6.81$ $\times 10^{-4}$ | $-4.03$ $\times 10^{-4}$ | $-5.79$ $\times 10^{-4}$ | $-4.99$ $\times 10^{-4}$ | $-4.28$ $\times 10^{-4}$ | $-5.15$ $\times 10^{-4}$ | $-2.54$ $\times 10^{-4}$ | $-1.56$ $\times 10^{-5}$ | $-4.40$ $\times 10^{-5}$ | $-1.50$ $\times 10^{-5}$ |

(d)   According to Equation (1), the loads except for 3 constraint points are calculated, as shown in Table 4.

**Table 4.** Fifteen points input loads of the first stage.

| Load | F (N) | Load | F (N) |
|---|---|---|---|
| $F_1$ | $-3270.36$ | $F_9$ | $-4575.70$ |
| $F_2$ | $-4906.32$ | $F_{10}$ | $-1985.09$ |
| $F_3$ | $-1768.35$ | $F_{11}$ | $-3243.11$ |
| $F_4$ | $-3035.01$ | $F_{12}$ | $-2657.57$ |
| $F_5$ | $-3131.87$ | $F_{13}$ | $-6133.72$ |
| $F_6$ | $-3965.12$ | $F_{14}$ | $-2611.90$ |
| $F_7$ | $-2610.04$ | $F_{15}$ | $-6017.92$ |
| $F_8$ | $-3992.43$ | - | - |

(e)   The force and moment balance equation of the whole system is as follows:

$$F_1(x_1 - x_G) + F_2(x_2 - x_G) + \cdots F_{15}(x_{15} - x_G) = 0 \tag{20}$$

$$F_1(y_1 - y_G) + F_2(y_2 - y_G) + \cdots F_{15}(y_{15} - y_G) = 0 \tag{21}$$

$$F_1 + F_2 + \cdots F_{15} + G = 0 \tag{22}$$

According to Equations (20)–(22), the loads of $F_3$, $F_{11}$ and $F_{15}$ can be solved. So far, all loads at 15 points have been calculated, as shown in Table 4.

### 3.2. Gravity Balance System Model

Fifteen points are grouped by a minimum distance of points for each unit. The first stage: two-point units are (1, 5), (2, 6) and (8, 11); three-point units are (3, 4, 7), (9, 12, 13) and (10, 14, 15). The first-stage balance point $1'$, $2'$, ... $6'$ can be obtained through Equations (11)–(13) and (17)–(19). The locations and loads of the first stage are shown in Table 5. The x-direction (height) coordinates of the first stage are unified as 1.3 m, which is conducive to the structural design of the gravity balance system. The gravity balance system is reduced from 15 points in the first stage to 6 points.

**Table 5.** The first stage of the gravity balance system.

| First Stage | Load Point | F′ (N) | X (m) | Y (m) | Z (m) |
|---|---|---|---|---|---|
| $1'$ | 1, 5 | $-6402.23$ | 1.3 | $-0.740$ | $-3.556$ |
| $2'$ | 2, 6 | $-8871.44$ | 1.3 | 0.091 | $-3.616$ |
| $3'$ | 3, 4, 7 | $-7413.4$ | 1.3 | 2.260 | $-3.698$ |
| $4'$ | 8, 11 | $-7235.54$ | 1.3 | $-0.554$ | $-0.647$ |
| $5'$ | 9, 12, 13 | $-13,367$ | 1.3 | 0.398 | $-0.287$ |
| $6'$ | 10, 14, 15 | $-10,614.9$ | 1.3 | 2.422 | $-0.111$ |

In the second stage, the six points are grouped. The two-point units are $(2', 3')$, $(1', 4')$ and $(5', 6')$. Finally, the locations and loads of the three-point $1''$, $2''$ and $3''$ are obtained. The x-direction (height) coordinates are unified as 1.4 m, as shown in Table 6.

The structure model of the multi-point gravity balance system is established according to the mathematical model, as shown in Figure 8.

**Table 6.** The second stage of the gravity balance system.

| Second Stage | First Stage | F″ (N) | X (m) | Y (m) | Z (m) |
|---|---|---|---|---|---|
| 1″ | 2′<br>3′ | −16,284.8 | 1.4 | 1.078 | −3.654 |
| 2″ | 1′<br>4′ | −13,637.8 | 1.4 | −0.641 | −2.012 |
| 3″ | 5′<br>6′ | −23,981.9 | 1.4 | 1.294 | −0.209 |

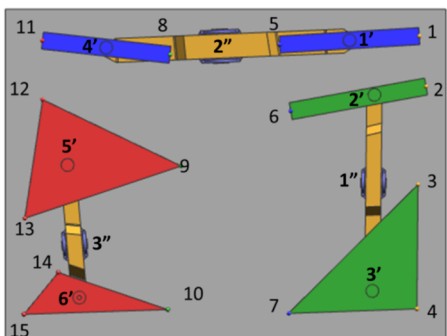

**Figure 8.** The multi-point gravity balance system. Point 1, 2, 3, 4 on the front frame; Point 5, 6, 7 on the first middle frame; Point 8, 9, 10 on the second middle frame; Point 11, 12, 13, 14, 15 on the main frame.

## 4. Analysis of the Influence of the Multi-Point Gravity Balancing System on the Imaging Quality Test

In order to achieve imaging quality testing, the absolute value of the displacement of the secondary mirror along the gravity direction is less than 160 μm. In this section, the displacements of the mirrors under the multi-point gravity balance system and the traditional support are analyzed in the ground imaging quality test. The finite element model of the large-aperture space camera and the gravity balance system is shown in Figure 9.

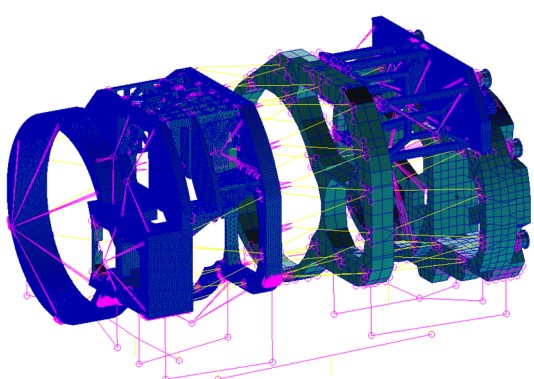

**Figure 9.** The finite element model of the large-aperture space camera and the gravity balance system.

### 4.1. Imaging Quality Analysis under the Traditional Support

In order to ensure the stiffness requirements and the capacity of the launch, there are multi-point connections between the satellite platform and the space camera. The connection points unlock until only three points remained in the orbit, which can reduce

the influence of the satellite platform stability on the camera. In the ground imaging quality test, the working state of the orbit is simulated. The displacement of each mirror is carried out under the three-point connection to the platform by the finite element analysis. The displacement in the gravity direction of the secondary mirror is able to reach −2650.87 μm. The absolute value of the displacement is more than 160 μm. The displacement nephogram of the space camera is shown in Figure 10, and the displacements of each mirror are shown in Table 7. The average wave front error of the full field of view of the optical camera (15 × 15, a total of 225 fields of view) is decreased from the optical design value of 0.027 $\lambda$ to 3.33 $\lambda$ ($\lambda$ = 632.8 nm), which could not realize the imaging quality test. The WFE is shown in Figure 11.

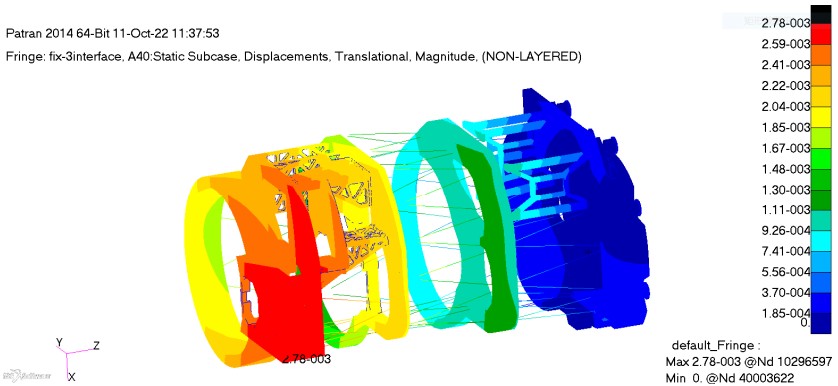

**Figure 10.** The displacement of the large-aperture space camera under the traditional support.

**Table 7.** The displacement of each mirror under the traditional support.

| Mirror | Tx (μm) | Ty (μm) | Tz (μm) |
|---|---|---|---|
| Primary mirror | −50.58 | −1.85 | −121.26 |
| Secondary mirror | −2650.87 | −151.58 | −171.41 |
| Tertiary mirror | −38.88 | −2.99 | −154.84 |

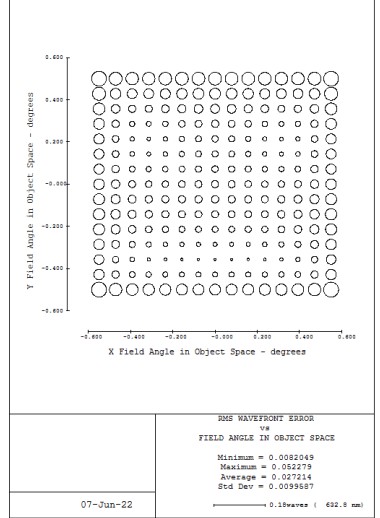
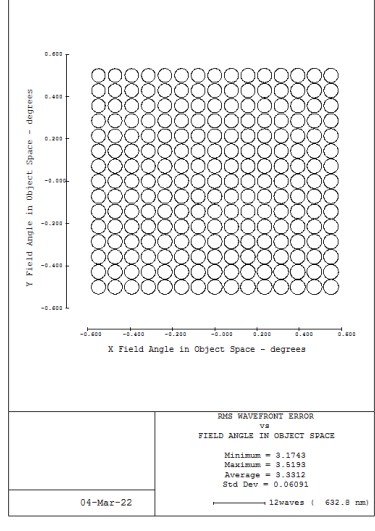

**Figure 11.** The WFE of the large-aperture space camera under the traditional support.

## 4.2. Imaging Quality Analysis under the Multi-Point Gravity Balancing System

In this section, the influence of the gravity balance system on the imaging quality test of a large-aperture space camera is analyzed. The displacement of each mirror is carried out under the gravity balance system by the finite element analysis. The displacement

in the gravity direction of the secondary mirror is −16.85 μm. The absolute value of the displacement is less than 160 μm. The displacement nephogram of the space camera is shown in Figure 12, and the displacements of each mirror are shown in Table 8. The average wave front error of the full field of view of the optical camera (15 × 15, a total of 225 fields of view) is 0.107 λ (λ = 632.8 nm). The WFE is 0.0367 λ through the secondary mirror pre-compensation, which can realize the imaging quality test. The WFE is shown in Figure 13.

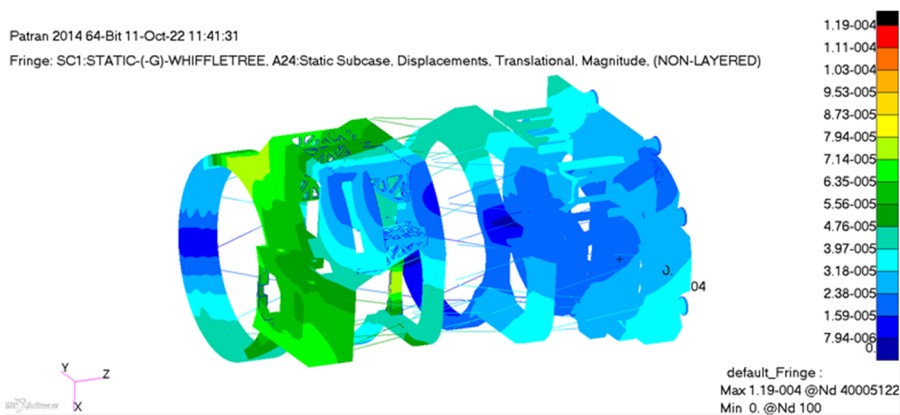

**Figure 12.** The displacement of the large-aperture space camera under the gravity balance system.

**Table 8.** The displacement of each mirror under the gravity balance system.

| Mirror | Tx (μm) | Ty (μm) | Tz (μm) |
|---|---|---|---|
| Primary mirror | −7.32 | −20.48 | 1.30 |
| Secondary mirror | −16.85 | 57.23 | −9.88 |
| Tertiary mirror | −4.55 | −17.89 | −25.90 |

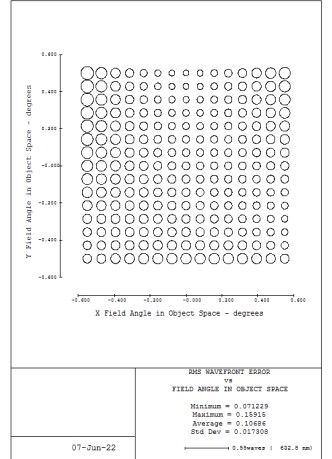 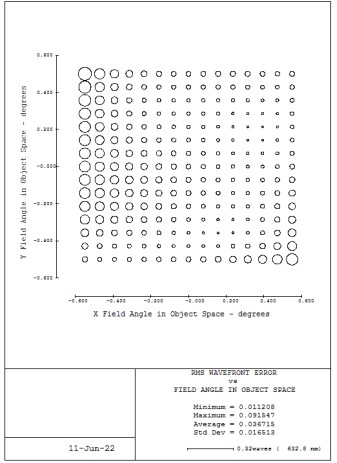

**Figure 13.** The WFE of the large-aperture space camera under the gravity balance system.

## 5. Discussion

In this paper, the load-solving method for each point of the multi-point system is proved. The mechanism and mathematical model of the multi-point gravity balance system was studied. According to the structure characteristics and load distribution of a large-aperture space camera, a multi-point gravity balance system model is designed. The influence on the imaging quality is analyzed. The WFE is 0.107 λ under the gravity balance system, which is much smaller than the traditional support. The multi-point gravity balance method is effective on the space camera imaging quality test. In the future, the multi-point

number and location of the gravity balance system can be optimized to further reduce the displacement of the mirror, which can realize a better imaging quality test.

**Author Contributions:** Conceptualization, L.Z.; Methodology, L.Z.; Validation, L.Z., M.A., X.L. and W.W.; Analysis, L.Z., M.A., X.L. and W.W.; Investigation, L.Z.; Resources, L.Z.; Data Curation, L.Z.; Writing—Original Draft Preparation, L.Z.; Writing—Review & Editing, L.Z. All authors have read and agreed to the published version of the manuscript.

**Funding:** This research was funded by National Natural Science Foundation of China: No. 11903034.

**Institutional Review Board Statement:** Not applicable.

**Informed Consent Statement:** Informed consent was obtained from all subjects involved in the study.

**Data Availability Statement:** The data presented in this study are available upon request from the corresponding author.

**Conflicts of Interest:** The authors declare no conflict of interest.

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
