# Peer review of "Research on the Multi-Point Gravity Balance Method for the Large-Aperture Space Camera in the Imaging Quality Test"

_photonics, doi:10.3390/photonics10070806_

Round 1
Reviewer 1 Report
The gravity balance system of the space camera with low stiffness and irregular mass distribution is innovatively proposed.The load solution method is derived.A mathematical model of the system is established. The gravity balance system established for a large aperture space camera which can realize imaging quality test.
1.Supplement the selection principle of 15-point gravity unloading of a large aperture space camera.
2. In the introduction, it is needed a brief introduction to the support of several telescopes.
3. It is recommended to carry out gravity balance optimization based on the best imaging quality in the future.
4. The “balacing ”in the first paragraph of Section 4 should be “balance”.
Minor editing of English language required.
Author Response
Thank you very much for your suggestions.The article has been revised according to your comments.Please see the attachment.

Reviewer 2 Report
This paper mentioned to be used for the imaging quality test of the large aperture space cameras with large weight and low stiffness. The paper gives the detailed introduction of multi-point gravity balance method. From the mathematical model of two-point unit and three-point unit, the relation between large weight with low stiffness and mathematical calculation is not clear, and this innovative part cannot be proved well. The displacement of each mirror under gravity balance system is given, but the quantitative results of influence on the imaging quality cannot be explained to show whether the displacement can meet the final imaging demands. The biggest issue is the lack of a more specific and detailed analysis of the proposed method to explain the relation between theory and application object with large diameter and low stiffness. The other aspect is the lack of experiment results to verify the calculation accuracy.
There are just a few minors and major notes that would improve clarity and provide more contexts.
Major comments:
l Please modify the paper to add the theory relation to prove the application object on large aperture and low stiffness.
l Please add the final experimental results to prove the calculation accuracy.
Minor revisions are recommended hereafter.
l The last sentence above equation 1 may change to “N displacements and forces equations are defined as follows”.
l Line 131 may be “point 3,11 and 15”.
l Discussion should be the detailed conclusion of the proposed methods and experiments to show the innovative parts using the exact results. It should be same with abstract.
For the above, I suggest the authors to make all the revisions and restructure the article so that it can be considered for publication.
Minor revisions are recommended hereafter.
l The last sentence above equation 1 may change to “N displacements and forces equations are defined as follows”.
l Line 131 may be “point 3,11 and 15”.
Author Response

(The authors gave the same response as above.)

Reviewer 3 Report
This paper presents a detailed analysis of the authors' multi-point gravity balance design. My feedback based on paper's sections re as follows:
1. Introduction
The article makes the case for a multi-point gravity balance method. However, only 2 references have been cited. While the author's do a good job pointing out the potential advantages of their methodology, no references are made to other prior designs for similar applications. For example, what other designs could be considered for high mass and low stiffness structures? How does the multi-point gravity balance method compare with them and what potential advantages could it have. The idea behind the authors' design needs to be sufficiently motivated.
2. Study on mathematical model of multi-point gravity balancing method:
It is not sufficiently clear how do the two point and three point models extend to the multi-point method. This section briefly explains the methodology, however no mathematical formulation is presented. It will be nice to see a few lines describing the method mathematically.
3. Multi-point gravity balance system model for a large aperture space camera.
The analysis is well described and sufficiently detailed.
4. Analysis of the influence of the multi-point gravity balancing system on the imaging quality test
The analysis is well described and sufficiently detailed.
General comments:
This paper provides a useful reference for the design of support structures for high accuracy optical applications specially in the case of high mass and low stiffness payloads. While most analysis presented here is static structural, it will also be interesting to see future work on how the multi-point balance structure handles transient loads. Overall this article provides a fairly detailed analysis that can be useful for other groups working on similar structures.
Minor spelling corrections and grammatical errors need to be checked and corrected.
Author Response

(The authors gave the same response as above.)
